# Repetitive Sequence Transcription in Breast Cancer

**DOI:** 10.3390/cells11162522

**Published:** 2022-08-14

**Authors:** Walter Arancio, Claudia Coronnello

**Affiliations:** Advanced Data Analysis Group, Fondazione Ri.MED, 90133 Palermo, Italy

**Keywords:** breast cancer, repetitive sequences, HERV, endogenous retrovirus, satellite repeats, centromeres, telomeres, SVA, LINE1, transposons

## Abstract

Repetitive sequences represent about half of the human genome. They are actively transcribed and play a role during development and in epigenetic regulation. The altered activity of repetitive sequences can lead to genomic instability and they can contribute to the establishment or the progression of degenerative diseases and cancer transformation. In this work, we analyzed the expression profiles of DNA repetitive sequences in the breast cancer specimens of the HMUCC cohort. Satellite expression is generally upregulated in breast cancers, with specific families upregulated per histotype: in HER2-enriched cancers, they are the human satellite II (HSATII), in luminal A and B, they are part of the ALR family and in triple-negative, they are part of SAR and GSAT families, together with a perturbation in the transcription from endogenous retroviruses and their LTR sequences. We report that the background expression of repetitive sequences in healthy tissues of cancer patients differs from the tissues of non-cancerous controls. To conclude, peculiar patterns of expression of repetitive sequences are reported in each specimen, especially in the case of transcripts arising from satellite repeats.

## 1. Introduction

### 1.1. Breast Cancer Classification

Breast cancer is still the leading cause of mortality among the female population in developed countries. In post-menopausal women, it accounts for 23% of all cancer deaths [1]. Breast cancers can be classified following anatomical, histological and molecular features [1], and their classification is a dynamic process, as stated in the last World Health Organization classification of tumors of the breast [2], and novel entities are added to the classification year by year following the increase in the knowledge of the disease [3]. Breast cancer is as a heterogeneous disease with different clinical and pathological features, variable therapeutic approaches and responses and with different outcomes even within the same class of breast cancer, suggesting that the current classifications are far from exhaustive. In order to follow the original classification of the cohort used in this study [4,5], the breast cancer specimens have been classified as: luminal-A, luminal-B (HER2-negative), luminal-B (HER2-positive), HER2-enriched and triple-negative breast cancers. This classification is based on immunohistochemical-relevant markers and was recommended by the St. Gallen Expert Consensus [6] and it has become a standard in routine clinical analysis since then [6,7,8]. For detail reviews, please refer to Hennigs et al. [8] and Prat et al. [9].

### 1.2. Non-Coding RNA in Mammals

The mammalian genome is pervasively transcribed and only a small portion of the transcriptional output has protein-coding potential [10]. The non-coding RNAs (ncRNAs) can be categorized using sizes and function, such as small-nuclear RNAs (snRNAs), small-nucleolar RNAs (snoRNAs), long non-coding RNAs (lncRNAs) and many others. The most well-studied class of ncRNAs is probably represented by microRNAs (miRNAs). Many studies have identified or suggested their role in human health and diseases, aging [11], cancer [12,13], diagnostic or prognostic purposes [14,15] and as therapeutic agents [16], either per se or in complex networks of cross regulation by the name of competing endogenous RNAs (ceRNAs) [17,18,19,20,21,22,23,24]. An abundant class of ncRNAs with variable functions that are still not fully understood is represented by transcripts arising from non-coding DNA sequences that are repeated along the genome in multiple copies. Even if the transcription from a single copy can be negligible, the sum of transcripts arising from thousands or millions of copies can be massive. A detailed description of them is given in the following paragraphs.

### 1.3. Repetitive DNA Sequence Classification

A multifaceted category of ncRNAs, of growing interest due to their roles in human health and diseases, is represented by the transcripts arising from repetitive DNA sequences (RS), i.e., DNA sequences that are present in multiple copies in the genomes, with low or nonexistent coding potential. RS represent about 45% of the human genome and are differentially transcribed in many tissues [25]. In mammals, RS have many roles in development and epigenetic regulation, but also in diseases such as cancer transformation [26,27,28,29,30] and degenerative diseases [31], but they are notoriously difficult to study [32]. Due to their nature, length and origin, RS can be roughly classified as: (i) Satellite repeats: a tandem array of simple or complex sequence repeats, abundant in heterochromatic regions, including alpha satellite repeats that represent the main DNA component of human centromeres. (ii) Long interspersed nuclear elements (LINEs): retrotransposons devoid of long terminal repeats (non-LTR) including some that are still able to retrotranspose. (iii) Small interspersed nuclear elements (SINEs): non-autonomous retrotransposons including the Alu elements in humans, which are often involved in genomic rearrangements. (iv) Integrated LTR retroviruses, mainly represented by the human endogenous retrovirus (HERV) families. (v) Additionally, the families of DNA transposons, that are usually not active in humans (Figure 1). The role of RS is starting to be properly understood. E.g., in the human brain LINE-1 retrotransposons are actively transcribed and mobilized and they are suggested to play a role in shaping the adult human brain [33], there is also a suggested role of RS in a model of aging of human brain [34].

### 1.4. Repetitive DNA Sequence and Cancer

Increased levels of heterochromatic repetitive satellite-coded RNAs in mammary glands induce breast tumor formation in mice, altering the BRCA1-associated protein networks that are required for the proper stabilization of DNA replication forks that in turn lead to genomic instability [35]. In humans, patients with breast cancer that express high levels of RNA derived from alpha satellite have an increased risk of developing multiple cancers [36].

It is known that LINE-1-encoded retrotranscription activity is widespread and its inhibition can reduce the rate of proliferation and promote the differentiation of breast cancer cells [37]. LINE-1 (and Alu) hypomethylation, suggesting an increased transcription in cancer cells and thus their mobilization, has been associated with the HER2-enriched subtype of breast cancer with worst prognosis [38,39,40]. In the transgenic mice of a well-defined model of breast cancer progression, LINE-1 is upregulated at a very early stage of tumorigenesis [41]. Indeed, the altered expression patterns of LINE-1-coded ORF1 and ORF2 proteins, with differences in overall patient survival, have been reported in invasive breast cancers [42]. In specific cases, pesticide exposure induces LINE-1 reactivation, suggesting the role of LINE transcription in pesticide-induced breast cancer progression [43], and MET-LINE-1 chimeric transcripts identify a subgroup of aggressive triple-negative breast cancers [44]. Overall, it has been suggested that LINE-1 may contribute to the origin or progression of breast cancers [45].

There are many reports regarding Alu and other SINE elements within or surrounding *BRCA1* and *BRCA2* genes essential to genomic rearrangements or genetic mutations leading to etiopathogenic, prognostic or predisposing mutations of breast cancers, both in somatic and germ lines [46,47,48,49,50,51]; indeed, the demethylation of Alu sequences may induce, at the same time, both transcription and rearrangements of Alu sequences. Thus, Alu transcription is a marker of increased susceptibility to Alu-mediated genomic rearrangement or genetic mutation at Alu sites. Looking for a direct effect of Alu transcription, it is noteworthy that heterogeneous nuclear ribonucleoprotein C (HNRNPC) is essential in breast cancer cell survival by inhibiting the double-stranded-RNA (dsRNA)-induced interferon response. Indeed, dsRNA in this setting is highly enriched in Alu sequences [52], suggesting that an overexpression of Alu sequences is characteristic of many breast cancers and may have lethal effects in cancer cells if not controlled.

There is significant evidence regarding the use of HERV-K-coded proteins as tumor markers and immunologic targets [52,53,54,55,56,57,58] and in influencing cancer stemness [59]. It has even been suggested that they could act as etiological agents [60,61]. Indeed, the expression of HERV-K is upregulated and associated with the basal-like breast cancer phenotype [62] and a HERV-derived long non-coding RNAs (namely, TROJAN) promotes triple-negative breast cancer progression [63]. HERV can directly contribute to cancer progression by activating the ERK pathway and inducing migration and invasion [64]; it has been even suggested that the activation of HERV-K may be essential for the tumorigenesis and metastasis of breast cancer [65]. Indeed, HERV-K-derived RNAs and antibodies against HERV-K-coded proteins are elevated in the blood of patients at an early stage of breast cancer [66].

DNA transposons are the less active and less well-studied class of RS in humans. Nevertheless, few reports suggest their role in breast cancer [67]; however, they were not investigated further. In addition, a mechanism of *BRCA1* mutation in three unrelated French breast/ovarian cancer families, that can be generated by an abortive integration of the human Tigger1 DNA transposon, has been postulated [68].

**Figure 1 cells-11-02522-f001:**
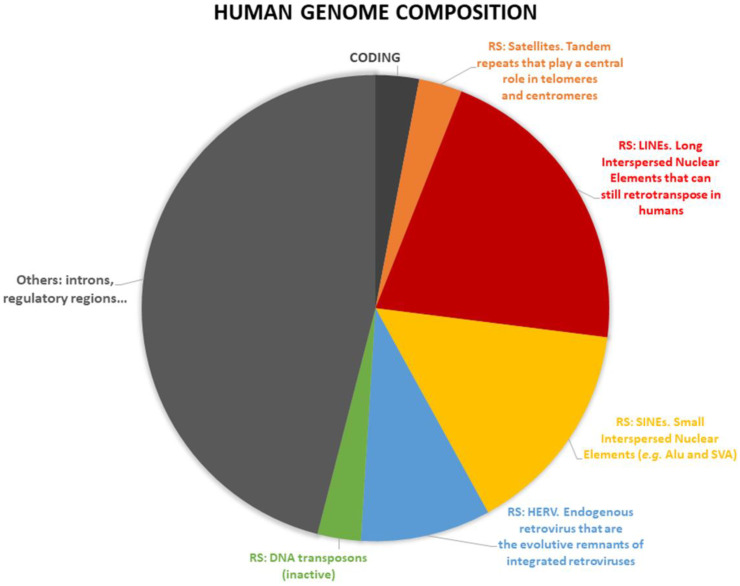
Repetitive sequences (RS) represent about half of the human genome. The panel reports RS activities associated with breast cancer. In orange: Satellite repeats [35,36]. In red: Long interspersed nuclear elements (LINEs) [37,42,45]. In yellow: Small interspersed nuclear elements (SINEs) [46,47,48,49,50,51]. In blue: Human endogenous retrovirus (HERV) [62,63,64,66]. In green: DNA transposons [68].

### 1.5. Main Aim

The main aim of this work was to analyze the transcripts arising from the repetitive sequences in a cohort of breast cancer specimens. Overall, we report peculiar patterns of expression and a diffuse upregulation of satellite transcription specific for each histotype.

## 2. Materials and Methods

### 2.1. Identifying and Quantifying the Repetitive Sequence Expression

The method of analysis and positive and negative controls (human beta actin cDNA NM_001101.4, 18S and 5.8 human ribosomal subunits, and the locus EF191515 of *Bacillus subtilis* SMY strain) have previously been described [34].

In brief, the analyses have been performed in a Galaxy environment [69,70]. The FASTQ raw sequences (obtained from the European Nucleotide Archive) have been uploaded, processed by Trimmomatic (Galaxy Version 0.36.5) [71] and quality checked. Bowtie2 aligner (Galaxy Version 2.3.4.2) [72] has been used with very sensitive local parameters to retrieve the expression of RS, oblivious of their genomic localization, aligning the reads against pseudochromosomes containing the reference RS sequences. ‘Very sensitive’ parameter takes into account the intrinsic sequence variability of RS, aiming to retrieve RS sequences that are slightly different from the canonical sequence used as reference. ‘Local’ alignment allows the retrieval of RS sequences attached to other sequences in the same reads, because RS are often embedded in other transcripts. Beta actin cDNA is a positive control of the pipeline used to retrieve the RS: RS are unspliced and the Bowtie2 aligner must efficiently retrieve the cDNA sequence of mature mRNAs, such as NM_001101.4, when treated as a pseudochromosome. Mature rRNA subunit sequences have been used as positive controls, following the same rational. Their large amount allows their retrieval even after rRNA depletion, which never reaches total efficiency, and they are physiologically unspliced, non-poly-adenylated ncRNAs of different lengths. Instead, EF191515 locus is a negative control that has no significant homology with human sequences and thus must have zero or almost zero reads in every sample. Raw data are reported in Appendix A.

### 2.2. Analysis of Coding Gene Expression

The raw FASTQ data have been aligned by the means of HISAT2 aligner (Galaxy Version 2.1.0) [73] using the Galaxy embedded hg38 as a reference. The generated BAM files have been compared using the hg38_GENCODE_GENE_V19.bed as a reference.

### 2.3. Statistical Analyses

The differential expression analysis was performed with the DESeq2 [74] algorithms implemented in a Galaxy environment (Version 2.11.40.2). The analysis was performed on the merged raw counts dataset, including coding genes, RS and control expression, if not otherwise specified. *p*-value correction for multiple comparisons was performed with the Benjamini and Hochberg method [75].

### 2.4. Dataset Used

We analyzed the data published in the European Nucleotide Archive (ENA), RRID:SCR_006515, study accession: PRJNA292118 [4,5]. The dataset contains sequencing data of 15 invasive breast cancer specimens (3 each for luminal-A, luminal-B (HER2-negative), luminal-B (HER2-positive), HER2-enriched and triple-negative breast cancer) and 18 controls (15 paired adjacent non-cancerous tissues and 3 healthy tissues). As reported in experiments SRX1135937 to SRX1135969 in the PRJNA292118 project [76], RNA was ribo-depleted via Ribo-Zero™ Gold Kits (human) before using the Illumina TruSeq RNA Sample Prep Kit (Cat#FC-122-1001) for the construction of the sequencing libraries. This kind of library allowed us to analyze both poly-adenylated and non-poly-adenylated transcripts; thus, it is suitable to analyze transcripts arising from RS, whose poly-adenylation status is often unknown.

## 3. Results

The expression of repetitive sequences has been retrieved and analyzed accordingly to the previously described pipeline [34] with minor adaptations. Raw expression data are reported in Appendix A. The normalized expression data of the merged raw counts of the coding genes and RS are reported in Appendix A. Whisker plots for selected RS of interest are reported in Figure 2.

### 3.1. Analysis of the Expression of Repetitive Sequences in Cancer Specimens

A comparison between the 15 invasive breast cancer specimens and 18 controls indicated a panel of RS differentially expressed in the two conditions (Table 1, Figure 2 and Appendix A). ALR, BSR and LSAU repetitive sequences are the most significantly upregulated, i.e., more than two-fold, in cancer specimens compared to the controls (P-adj < 0.05). However, the specimens showed great variability (Appendix A); thus, a comparison between each histotype with its adjacent normal tissues (ant) was performed.

In ‘HER2-enriched’ breast cancer, an upregulation of HSATII satellite expression is reported (Table 1; Appendix A). Regarding ‘luminal-A’ histotype, the expression of ALR is strongly upregulated (Table 1; Appendix A). In the case of ‘luminal-B HER2-negative’, the expression of ALR satellite family and “6 kb tandem repeat sequence in Homo sapiens” is upregulated (Table 1; Appendix A). We also report a striking upregulation of BSR satellite sequences between a specific luminal-B HER2-negative specimen and its adjacent non-cancerous tissues (namely, LUM_B_Her2_NEG_0). Its normalized expression is 208203 reads in the cancer specimen against 88 in the adjacent tissue (Appendix A). Thus, this specimen has the worst classification in TNM staging in the cohort (IIIb, together with another specimen) and many lymph node metastases (36 positives out of the 42 inspected). Analyzing ‘luminal-B HER2-positive’ specimens, we report a generalized upregulation of several satellite-derived transcripts together with a downregulation of endogenous retroviruses and their LTR sequences (Table 1; Appendix A). In ‘triple-negative breast cancer’, there is an upregulation of satellite sequences and a perturbation in the transcription of endogenous retroviruses and their LTR sequences (Table 1; Appendix A).

### 3.2. Analysis of Expression Background

A detailed analysis of the comparison between the expression of ANT and the normal controls is reported in Table 2 and Appendix A. The most differentially expressed RS is MER22 satellite (also called SST1) [77,78], which is downregulated in ANT compared to the controls. Another abundant RS that is differentially expressed in ANT compared to the controls is SVA_A, which is also downregulated. SVAs are SINEs that contain Alu sequences. SVAs are still active in humans and may have biological effects [79].

We also report that the vast majority of the specimens analyzed showed an altered expression of GGAAT repeats in comparison with their specific ANT, either increased (GGAAT^higher^) or decreased (GGAAT^lower^).

## 4. Discussion

High-throughput RNA sequencing helps one analyze the pervasive transcription of the human genome, but it bears the burden of processing a huge amount of data. In the routinely used pipelines of analysis, the transcription of RS is often overlooked due to the intrinsic difficulties to be analyzed by the most common means. Nevertheless, RS represent about half of the human genome and the source of a fair amount of transcriptional output. Indeed, whenever analyzed, transcripts arising from RS showed biological and medical properties far beyond the role of simple bystanders or byproducts [31,32].

While several studies on breast cancers highlighted a potential role of specific RS as etiopathogenic agents or as diagnostic or prognostic tools, this is the first study analyzing the expression of RS as a whole in a panel of breast cancer specimens classified by their molecular characteristics.

Overall, it is evident that the cancerous specimens showed an increased expression from satellite repeats, suggesting centromeric and telomeric loss of heterochromatinization and thus chromosomal instability [80,81]. In particular, it has been previously demonstrated that the overexpression of alpha satellite transcripts leads to chromosomal instability in breast cancer via segregation errors [82].

It is interesting that each histotype and, more generally, each specimen showed a specific altered pattern of transcripts arising from RS and in detail from satellite repeats, suggesting that the altered transcription of RS could be something more than an epiphenomenon and may indicate the peculiar characteristic of the specimens, such as an effect of a previous viral infection [83]. The specificity of RS transcription is supported by a case of ‘luminal-B HER2-negative’ in which the transcripts derived from BSR (beta satellite repeats) are thousands of times more upregulated in comparison with its background (Figure 2), a unique case among all the cases analyzed in this paper and others [84]. It is noteworthy that this case has the worst classification in TNM staging and diffuse lymph node metastases. The increase may be due to a true increase in BSR expression or a significant increase in BSR-derived transcript polyadenylation and stability. Indeed, satellite polyadenylation has been postulated in humans, following studies on other organisms [85], and evidence is now arising and being consolidated [86]. The analysis of RS transcription suggests that the current molecular classification of breast cancers, even if functional in defining the therapeutic course [9], is far from being exhaustive in defining their molecular characteristics. This is in line with the great variability of prognosis and clinical course in the same class of breast cancers [86].

We also report that the vast majority of the specimens analyzed showed an altered expression of GGAAT repeats in comparison with their specific ANT, either increased (GGAAT^higher^) or decreased (GGAAT^lower^). The GGAAT^higher^ specimens have a higher ki67 staining [87], a widely used marker of cell proliferation, pointing to a faster growing tumor mass. Indeed, the two specimens that had the worst TNM classification (IIIb) and an evident lymph node involvement are GGAAT^higher^.

Regarding the transcriptional background, comparing the healthy adjacent tissues of cancer specimens with tissues from healthy donors, there is a generalized downregulation of MER22 satellite expression. MER22 has been implied in meiotic instability [76,88], and its methylation status has been found relevant to multiple cancer types [89,90,91,92].

Interestingly, ALR satellites are upregulated in ANTs. Considering that ALR expression is also upregulated in tumors in comparison with non-cancerous specimens, this suggests the role of these satellites in the progression of the disease.

Overall, the altered transcriptional landscape of RS in the background of patients may suggest either a genetic predisposition and increased susceptibility to cancer transformation or it could be the result of epigenetic alteration due to environmental factors (e.g., exposition to chemicals or previous infections), which ultimately need to be investigated further.

## 5. Conclusions

The analysis of transcripts arising from RS in breast cancer specimens classified following the current molecular markers showed a great interspecimen variability, with peculiar patterns of altered RS transcription. Overall, there is an evident alteration in the transcripts arising from satellite repeats and, in specific cases, from SINE and endogenous retrovirus sequences. The expression from healthy adjacent tissues of cancer specimens showed an altered expression of RS transcription when compared to the transcription of healthy donors. If the data presented here are confirmed and extended in a larger population, RS expression may play a role in the molecular classification and stratification of patients or even be potentially adopted as a biomarker in liquid biopsy.

## Figures and Tables

**Figure 2 cells-11-02522-f002:**
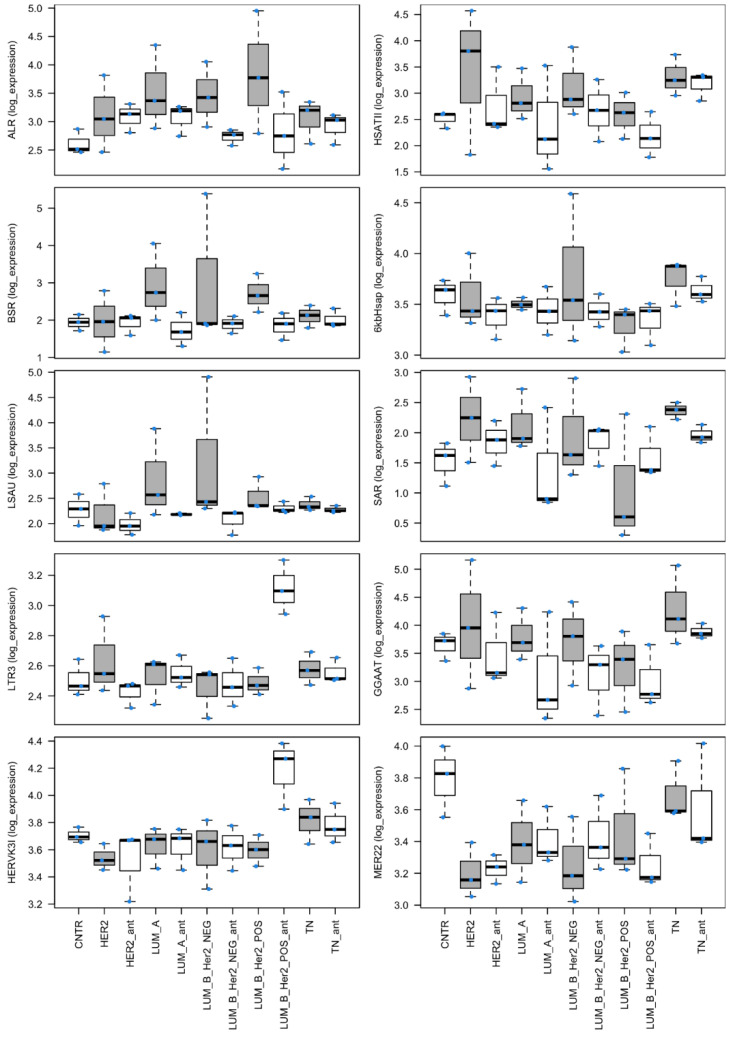
Whisker plots of the expression of selected RS in Log10. The cancerous specimens are plotted in gray.

**Table 1 cells-11-02522-t001:** A comparison between the 15 invasive breast cancer specimens and 18 controls (cancer vs. normal) and between each histotype with its adjacent non-cancerous tissues highlighted a panel of RS differentially expressed in the two conditions. The RS whose mean expression is above 100 and with a *p*-value < 0.05 are reported.

GeneID	Base Mean	log2(FC)	StdErr	Wald-Stats	*p*-Value	P-adj
Cancer vs. normal						
BSR	597.99	1.62	0.28	5.80	6.57 × 10^−9^	1.93 × 10^−4^
ALR	5004.62	1.52	0.28	5.44	5.24 × 10^−8^	7.70 × 10^−4^
LSAU	484.17	1.36	0.28	4.91	9.31 × 10^−7^	8.20 × 10^−3^
ALRb	1358.44	1.03	0.27	3.81	1.37 × 10^−4^	1.44 × 10^−1^
ALR1	4308.91	0.99	0.27	3.73	1.90 × 10^−4^	1.48 × 10^−1^
GGAAT	11,375.95	0.91	0.28	3.28	1.05 × 10^−3^	3.35 × 10^−1^
HSATII	2167.60	0.91	0.28	3.27	1.08 × 10^−3^	3.35 × 10^−1^
PABL_BI	121.89	0.46	0.16	2.89	3.86 × 10^−3^	5.15 × 10^−1^
SAR	137.70	0.75	0.28	2.69	7.08 × 10^−3^	6.09 × 10^−1^
LTR22B2	517.80	−0.32	0.13	−2.52	1.17 × 10^−2^	7.00 × 10^−1^
LTR72	226.51	−0.30	0.12	−2.41	1.58 × 10^−2^	7.74 × 10^−1^
LTR12C	39,355.33	−0.51	0.22	−2.36	1.84 × 10^−2^	8.01 × 10^−1^
MER9B	648.51	−0.26	0.12	−2.27	2.33 × 10^−2^	8.33 × 10^−1^
LTR7B	2761.48	−0.35	0.16	−2.21	2.70 × 10^−2^	8.50 × 10^−1^
LTR7C	483.73	−0.25	0.12	−2.11	3.46 × 10^−2^	8.83 × 10^−1^
LTR35	156.02	−0.21	0.10	−2.10	3.61 × 10^−2^	8.86 × 10^−1^
ALR_	10,312.15	0.49	0.24	2.00	4.51 × 10^−2^	9.16 × 10^−1^
HER2 vs. ant						
ZAPHOD	160.72	−0.86	0.41	−2.07	3.81 × 10^−2^	1.00
HSATII	6250.25	1.10	0.56	1.96	4.98 × 10^−2^	1.00
LumA vs. ant						
ALR	5513.55	1.11	0.44	2.52	1.16 × 10^−2^	1.00
LTR38B	472.48	−1.04	0.45	−2.32	2.04 × 10^−2^	1.00
LumB_Her2Neg vs. ant						
ALR	2804.92	1.75	0.48	3.64	2.74 × 10^−4^	1.29 × 10^−1^
ALR1	3720.86	1.36	0.46	2.96	3.07 × 10^−3^	3.40 × 10^−1^
ALRb	1210.79	1.13	0.45	2.52	1.19 × 10^−2^	6.05 × 10^−1^
MER57C1	196.12	1.00	0.43	2.31	2.08 × 10^−2^	7.23 × 10^−1^
6kbHsap	7701.13	1.06	0.49	2.18	2.94 × 10^−2^	8.01 × 10^−1^
LumB_Her2Pos vs. ant						
LTR3	853.33	−1.87	0.45	−4.14	3.45 × 10^−5^	1.33 × 10^−2^
HERVK3I	10,531.29	−1.76	0.47	−3.72	1.99 × 10^−4^	4.53 × 10^−2^
BSR	436.11	2.09	0.60	3.49	4.92 × 10^−4^	8.70 × 10^−2^
ALR1	20,223.11	2.06	0.64	3.21	1.32 × 10^−3^	1.56 × 10^−1^
ALR	14,122.78	1.97	0.64	3.06	2.24 × 10^−3^	2.10 × 10^−1^
LTR1B0	5449.61	−1.94	0.64	−3.02	2.50 × 10^−3^	2.23 × 10^−1^
LTR12C	61,931.23	−1.35	0.55	−2.45	1.45 × 10^−2^	5.91 × 10^−1^
MER122	128.05	1.31	0.64	2.06	3.97 × 10^−2^	9.10 × 10^−1^
ALR_	13,301.52	1.18	0.59	1.99	4.61 × 10^−2^	9.55 × 10^−1^
TN vs. ant						
SAR	171.13	0.84	0.28	3.01	2.63 × 10^−3^	2.41 × 10^−1^
LTR72B	423.96	−0.56	0.22	−2.57	1.03 × 10^−2^	4.98 × 10^−1^
MER87B	580.38	0.43	0.19	2.23	2.59 × 10^−2^	7.70 × 10^−1^
GSAT	291.10	0.58	0.29	1.98	4.72 × 10^−2^	9.7 × 10^−1^

**Table 2 cells-11-02522-t002:** Comparison between the expression of RS in ANT and the normal controls (ANT vs. normal). The RS whose mean expression is above 100 and with a *p*-value < 0.05 are reported.

GeneID	Base Mean	log2(FC)	StdErr	Wald-Stats	*p*-Value	P-adj
MER22	3245.14	−1.03	0.35	−2.95	3.18 × 10^−3^	6.42 × 10^−1^
PABL_BI	101.04	0.67	0.24	2.82	4.78 × 10^−3^	6.42 × 10^−1^
PTR5	394.09	−1.10	0.41	−2.67	7.67 × 10^−3^	6.42 × 10^−1^
TAR1	184.59	−0.66	0.26	−2.53	1.15 × 10^−2^	6.42 × 10^−1^
LTR7C	516.36	−0.55	0.23	−2.44	1.45 × 10^−2^	6.42 × 10^−1^
GSAT	178.69	−1.22	0.51	−2.37	1.79 × 10^−2^	6.42 × 10^−1^
LTR46	249.63	−0.69	0.31	−2.18	2.94 × 10^−2^	6.42 × 10^−1^
LTR7Y	1511.10	−0.76	0.35	−2.16	3.05 × 10^−2^	6.42 × 10^−1^
LTR1E	413.79	−0.59	0.28	−2.15	3.19 × 10^−2^	6.42 × 10^−1^
LTR7A	2067.24	−0.77	0.36	−2.14	3.20 × 10^−2^	6.42 × 10^−1^
LTR22B2	563.94	−0.50	0.23	−2.14	3.27 × 10^−2^	6.42 × 10^−1^
LTR44	116.12	0.81	0.39	2.08	3.79 × 10^−2^	6.42 × 10^−1^
LTR27C	315.21	−0.59	0.28	−2.08	3.79 × 10^−2^	6.42 × 10^−1^
ALR	1028.61	1.00	0.48	2.07	3.83 × 10^−2^	6.42 × 10^−1^
HARLEQUINLTR	3541.69	−0.45	0.22	−2.07	3.84 × 10^−2^	6.42 × 10^−1^
MER51C	493.52	−0.45	0.22	−2.03	4.19 × 10^−2^	6.42 × 10^−1^
SVA_A	65,198.32	−0.57	0.28	−2.02	4.32 × 10^−2^	6.42 × 10^−1^
MER54B	165.03	−0.57	0.28	−2.00	4.54 × 10^−2^	6.42 × 10^−1^

## Data Availability

The analyzed data are published in the European Nucleotide Archive (ENA), RRID:SCR_006515, study accession: PRJNA292118 [4,5].

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
