# Peer review of "Repetitive Sequence Transcription in Breast Cancer"

_cells, 2022, doi:10.3390/cells11162522_

Round 1
Reviewer 1 Report
Repetitive sequences represent about half of the human genome. They have been shown to be actively transcribed and play a role during development and in epigenetic regulation. The altered activity of repetitive sequences can lead to genomic instability and they can contribute to the establishment or the progression of degenerative diseases and cancer transformation. In this manuscript b,Arancio and Coronnello the authors have analyzed the expression profiles of DNA repetitive sequences in the breast cancer specimens. Satellites expression is generally upregulated in breast cancers, with specific families upregulated per histotype: in HER2-enriched cancers they are the human satellite II (HSATII); in Luminal A and B they are part of the ALR family and in triple negative SAR and GSAT, together with a perturbation in the transcription from endogenous retroviruses and their LTR sequences. Interestigly, the authors repotred that the background expression of repetitive sequences in healthy tissues of cancer patients differs from tissues of non-cancerous controls. Thus, the authors demonstrated peculiar patterns of expression of repetitive sequences in specimens, especially in the case of transcripts arising from satellite repeats.
Although scientificallyi the manuscript is on sound footing, it can be improved by including figures or diagrams that better explain role repetitive sequences , through few concrete examples.
Author Response
We thank the reviewer for these suggestions. Figure 1 was added in the introduction and figure 2 was added in the results, in order to improve the readability of the manuscript.
Reviewer 2 Report
Review on the manuscript titled “Repetitive Sequences Transcription in Breast Cancer” by Arancio, Coronnello, 2022.
The authors addressed the expression profiles of repetitive sequences (rs) in breast cancer (bc) cohorts published previously by other authors in ENA (PRJNA292118). The dataset contains sequencing data of 15 invasive breast cancer specimens (3 each for luminal-A, luminal-B (HER2-negative), luminal-B (HER2-positive), HER2-enriched and triple negative breast cancer) and 18 controls (15 paired adjacent non-cancerous tissues and 3 healthy tissues). Overall 5 bc types and not sure how many distinct controls.
The authors reassembled the raw data stressing their attention on rs. They report “Satellites expression is generally upregulated in breast cancers, with specific families upregulated per histotype: in HER2-enriched cancers they are the human satellite II (HSATII); in Luminal A and B they are part of the ALR family and in triple negative SAR and GSAT, together with a perturbation in the transcription from endogenous retroviruses and their LTR sequences. Noteworthy, we report that the background expression of repetitive sequences in healthy tissues of cancer patients differs from tissues of non-cancerous controls.”
While there are multiple instances of rs analysis in RNA-seq data published to date, the key is methodological issues with their assessments. As long as the majority of RNA-seq data use poly-A associated protocol, we cannot be sure that the expression assessments of rs are valid. The authors referenced the work [34] for the protocol used, but I’m still not sure it would provide valid results. E.g, the same is valid for miRNA: though RNA-seq data does contain expression evaluation for some of them (as well as quite a lot of rRNA sequences), it’s quite recommended using special kits for miRNA adequate detection/evaluation.
Thus, I’d like to underline several comments.
1) The manuscript should contain explicit description of rs capture protocol. Usually rs are blocked with most of mappers as ambiguously mapped. Lack of polyA tail jeopardizes process as well.
2) The authors should also justify/elaborate their problem statement in terms of assessments. In particular, it is obvious that satellite repeats express due to spurious transcription/chromatin availability. It could result from chromosome rearrangements/breaks. Did the authors try retrieving caryotypes of cells they study?
3) The point that Cancer types vs ANT comparison yields significance in a range of cases (Table 1) might underlie choromosome rearrangements/breaks in cancerous tissues, which would strongly affect the problem statement/conclusions using the data.
4) Just 3 satellites (ALR, BSR, LSAU) were found significantly deviated from controls (Table 1). I assume there are several samples with abnormal quantity of these ones implying chromosome aberrations therein. It would be good to see whiskey plot across the bs samples for them, or better for all those significant in Table 1.
5) L1 transposon is autonomous unit with ability to retropose once getting accessible, but usually it is heterochromised. Chromosome aberrations may make these accessible as well. Alternatively, at least 10% of human coding genes contains ALU (SINE) fragments. That must be taken into account in analysis.
6) The chapter with ncRNA is out of relevance, I think its not relevant, and may be omitted.
7) Supplementary Table 1 is not of .csv type, but .tsv one.
Overall, lack of transparent methodology description with lack of possible background mechanisms of rs expression presented, along with possibly aberrant caryotypes within several samples, may greatly influence the research protocol validity.
Author Response
The manuscript should contain explicit description of rs capture protocol. Usually rs are blocked with most of mappers as ambiguously mapped. Lack of polyA tail jeopardizes process as well.
- The dataset used derived from libraries that were ribodepleted and NOT poly-A enriched. So the transcriptome sequenced contained all the RNAs without only rRNA (in an ideal condition) comprehensive of RS transcripts. These datasets have been ad hoc selected for this purpose from those present in public databases. The work 34 in reference describes in details the bioinformatic pipeline used to retrieve the information about the abundance of RS in the samples (paragraphs 2.1 and 2.3). In the methods section, the information has been added if missing (in 2.4) or detailed if too synthetic (in 2.1), and now it goes much more into the details of the pipeline and its rationale.
The authors should also justify/elaborate their problem statement in terms of assessments. In particular, it is obvious that satellite repeats express due to spurious transcription/chromatin availability. It could result from chromosome rearrangements/breaks. Did the authors try retrieving caryotypes of cells they study?
- That would be of extreme interest. Unlucky this is a silico research where we are analyzing raw data previously published and shared by other laboratories. Thus, we have not access to patients, samples or unpublished data.
The point that Cancer types vs ANT comparison yields significance in a range of cases (Table 1) might underlie choromosome rearrangements/breaks in cancerous tissues, which would strongly affect the problem statement/conclusions using the data.
- This was also our interpretation of data as g. stated in discussion “It is evident that overall the cancerous specimens showed an increased expression from satellite repeats, suggesting centromeric and telomeric loss of heterochromatinization and thus chromosomal instability [79,80]. In particular, it has been previously demonstrated that overexpression of alpha satellite transcripts leads to chromosomal in-stability in breast cancer via segregation errors [81]."
Just 3 satellites (ALR, BSR, LSAU) were found significantly deviated from controls (Table 1). I assume there are several samples with abnormal quantity of these ones implying chromosome aberrations therein. It would be good to see whiskey plot across the bs samples for them, or better for all those significant in Table 1.
- Thank you for your suggestion. Figure 2 was added accordingly. We think this would greatly improve the readability of the paper.
L1 transposon is autonomous unit with ability to retropose once getting accessible, but usually it is heterochromised. Chromosome aberrations may make these accessible as well. Alternatively, at least 10% of human coding genes contains ALU (SINE) fragments. That must be taken into account in analysis.
- This is one of the main problems working with RS. We are interested in the global effects of RS transcription, and our pipeline allows to retrieve differential RS expression oblivious of the genomic localization (this statement was added in the methods, 2.1). For example, in another work we found a specific SINE overexpressed in a medical condition under investigation (unpublished data, manuscript in preparation) and thus in that case we investigated the genes in whose it was embedded. I this paper instead we found significant differences mainly in satellite expression that had been widely associated with centromeres and (to a lesser degree) telomeres stability. For these reasons we did not extended the analysis to the gene level.
The chapter with ncRNA is out of relevance, I think its not relevant, and may be omitted.
- We prefer to retain this chapter because RS are in large part ncRNAs. A lot of readers are absolutely not confident with RS biology, but probably they know about ncRNAs. Presenting RS as a part of ncRNAs can help the readers to understand the topic under investigation. A small paragraph has been added to highlight that RS can be considered as abundant ncRNAs with variable functions, still not fully understood (1.2 and 1.3).
Supplementary Table 1 is not of .csv type, but .tsv one.
- We will amend and upload the file with the correct extension.
Overall, lack of transparent methodology description with lack of possible background mechanisms of rs expression presented, along with possibly aberrant caryotypes within several samples, may greatly influence the research protocol validity.
- We hope to have clarified the methodology. Instead, the aberrant karyotypes can be at the very root of the observations. We think that, if the data will be confirmed or extended in a larger datasets, satellite expression could be used for the molecular classification and stratification of patients or even be potentially adopted in liquid biopsy. A sentence has been added in the conclusion.
Reviewer 3 Report
Arancio and Coronnello investigated the expression of repetitive sequences in different breast cancer samples of the HMUCC cohort. The authors re-analysed previously published ribo-depleted RNA-seq samples and associated paired adjacent non-cancerous tissues. These samples include luminal-A, luminal-B HER-2 negative, luminal-B HER-2 positive, and HER-2 enriched and triple negative breast cancer. The authors found a general upregulation of satellites expression in breast cancer and differences in expression of some repetitive sequences between the adjacent non-cancerous tissues compared to healthy controls or breast cancer samples.
Comments:
1. The authors should describe in more details their Methods section rather than citing a previous paper. What are the functions in the RNA-seq analysis of each of the controls (beta actin, 18S, 5.8S, EF191515) the authors are using?
2. As some of the repetitive sequences can be located in 5’ or 3’UTRs, did the authors control for possible effects of changes in gene expression for these intragenic repetitive sequences? Downregulation or upregulation of a gene containing repetitive sequences in its 3’UTR would result in a downregulation/upregulation of these repetitive sequences if the expression of the gene is not considered.
3. The English will have to be improved throughout the manuscript.
Author Response
- The authors should describe in more details their Methods section rather than citing a previous paper. What are the functions in the RNA-seq analysis of each of the controls (beta actin, 18S, 5.8S, EF191515) the authors are using?
- We described more in details the method and added the rationale behind controls.
- As some of the repetitive sequences can be located in 5’ or 3’UTRs, did the authors control for possible effects of changes in gene expression for these intragenic repetitive sequences? Downregulation or upregulation of a gene containing repetitive sequences in its 3’UTR would result in a downregulation/upregulation of these repetitive sequences if the expression of the gene is not considered.
- This is one of the main problems working with RS. In this paper we are interested in the global effects of RS transcription, and our pipeline allows to retrieve differential RS expression oblivious of the genomic localization (this statement was added in the methods, 2.1). For example, in another work we found a specific SINE overexpressed in a medical condition under investigation (unpublished data, manuscript in preparation) and thus in that case we investigated the genes in whose it was embedded. I this paper instead we found as main result a significant difference in satellite expression that had been widely mainly associated with structural chromosomal parts such as centromeres and (to a lesser degree) telomeres. For these reasons we did not extended the analysis to the gene level.
- The English will have to be improved throughout the manuscript.
- We checked the manuscript and improved the English.
Round 2
Reviewer 1 Report
The manuscript has been sufficiently modified and the reviewer feels it is worthy enough for publication in the Journal.
Reviewer 2 Report
The authors addressed my notes in an acceptable form.